# CPAP Adherence Improvement for OSA Patients through Integrated Feedback Systems

Yasaman KAKAEI SIAHKAL
*Université Lumière Lyon 2, INSA Lyon,*
*Université Claude Bernard Lyon 1,*
*Université Jean Monnet Saint-Etienne,*
*DISP UR4570*
Bron, 69676, France
*Linde HomeCare*
Bourg en Bresse, France
yasaman.kakaei-siahkal@univ-lyon2.fr

Nejib MOALLA
*Université Lumière Lyon 2, INSA Lyon,*
*Université Claude Bernard Lyon 1,*
*Université Jean Monnet Saint-Etienne,*
*DISP UR4570*
Bron, 69676, France
Nejib.Moalla@univ-lyon2.fr

Aïcha SEKHARI
*Université Lumière Lyon 2, INSA Lyon,*
*Université Claude Bernard Lyon 1,*
*Université Jean Monnet Saint-Etienne,*
*DISP UR4570*
Bron, 69676, France
aicha.sekhari@univ-lyon2.fr

Tao WANG
*Université Jean Monnet Saint-Etienne,*
*INSA Lyon, Université Lumière Lyon 2,*
*Université Claude Bernard Lyon 1,*
*DISP UR4570*
Roanne, 42300, France
tao.wang@univ-st-etienne.fr

Olivier GRASSET
*Linde Homecare France*
Bourg-en-Bresse, France
olivier.grasset@linde.com

*Abstract*—**This study presents a groundbreaking strategy for the homecare management of Obstructive Sleep Apnea (OSA). With an emphasis on patient empowerment through the integration of feedback management systems into sleep treatment, this study offers an innovative approach to the homecare management of obstructive sleep apnea (OSA). Fundamentally the research attempts to significantly increase, via tailored interventions, adherence to Continuous Positive Airway Pressure (CPAP) therapy. It accomplishes this by combining qualitative patient feedback with quantitative CPAP machine monitoring data to improve patient clustering and in turn treatment outcomes. The research methodology is comprehensive, encompassing various stages that include advanced patient grouping, continuous incorporation of new patient data, integration of feedback from surveys on both intervention and medical sleep, and a thorough cycle of interventions and evaluations. This iterative refinement process is essential as it allows for the dynamic updating of patient profiles and clustering based on evolving data and treatment responses. All these efforts are focused on fostering a more tailored approach to patient care. The creation of patient-centered treatment plans that maximize treatment efficacy by utilizing intervention repositories and data analytics is at the heart of this research. The study also explores how personalized care can improve CPAP adherence and underscores the need to tailor interventions and content based on patient feedback. This multi-layered approach aims at improving patient treatment adherence. It creates a more efficient patient-centered care model for individuals with OSA by continuously adapting and personalizing the interventions provided to each patient, thereby fostering a stronger relationship between patients and their treatments.**

*Keywords—Patient feedback, data analytics, patient-centric treatment, intervention repositories, treatment efficiency design, Obstructive Sleep Apnea (OSA), Continuous Positive Airway Pressure (CPAP), CPAP Adherence, Personalized Care.*

## I. INTRODUCTION

Obstructive Sleep Apnea (OSA) is a common health problem, which affects individuals' health-related quality of life (HRQL) and sleep. It is explained as a chronic disease with repeatable respiratory pauses during sleep, accompanied by episodic hypoxia and sleep fragmentation[1]. The primary treatment for OSA is continuous positive airway pressure (CPAP), which normalizes the apnea-hypopnea index (AHI) and improves sleep quality, daytime sleepiness, and quality of life[2]. The paper highlights the development of the feedback management system intended to smoothly incorporate patient input into the therapy process as a novel way to manage obstructive sleep apnea (OSA) in a homecare setting. Our approach combines quantitative data from CPAP machine monitoring with qualitative data from monitoring surveys and medical sleep surveys, which evaluate broader sleep health and quality issues. Qualitative data for our study derived from two primary sources: first, interventions that are informed by monitoring survey that gathers patient feedback on their treatment experiences following each intervention; and second, medical Sleep Surveys to evaluate broader issues related to sleep health and quality. By integrating these qualitative observations with quantitative data, our approach emphasizes real-time feedback updates to continuously enhance CPAP adherence and patient engagement from initial diagnosis through continued management[3]. This patient-centric approach, which includes personalized care interventions based on patient profiling, has been shown to significantly improve adherence rates and overall health quality of life[4]. The impact of integrating these qualitative observations with quantitative data to inform and enhance patient care knowledge-based approach that emphasizes the development of a patient-centric treatment plan is highlighted in[5].

We plan to look closely at how patients adhere to their treatment and how they interact with the various stages of their therapy with the help of monitoring surveys, medical sleep surveys, and Monitoring CPAP devices. Additionally, based on the work of [5] insight us to consider how personalized care interventions based on patient profiling can enhance treatment adherence. By utilizing personalized interventions and advanced data clustering techniques, we can refine patient profiles and tailor intervention strategies more effectively, enhancing adherence[6].This approach is designed to tailor

treatment plans precisely to individual needs. Through this, we accomplished a more patient-centered approach that effectively supports individuals adhering to their prescribed therapy.

## II. STATE OF ART

### A. Research Domains

Our study focuses on three key fields to improve CPAP therapy for Obstructive Sleep Apnea (OSA), with consideration of patient and provider insights with adherence analysis. We aim to deeply understand the patient journey through CPAP therapy, finding their challenges and preferences with qualitative interviews and quantitative data. This approach helps us pinpoint where patient experiences and objectives align or diverge, enriching our strategy for patient-centered care[5],[7]. Recent studies have highlighted the importance of integrating behavioral interventions with CPAP therapy. These interventions, such as cognitive behavioral therapy for insomnia (CBT-I), have been shown to significantly enhance adherence to CPAP by addressing both the psychological and behavioral aspects of OSA treatment[8]. Moreover, integrating digital feedback mechanisms tailored to patient-specific CPAP usage patterns and concerns has been demonstrated to significantly improve adherence rates[9].

To further enhance patient adherence to CPAP therapy, we implement a multi-faceted approach that integrates CPAP monitoring, intervention monitoring surveys, and medical surveys. This methodology aims to create a comprehensive and personalized treatment plan by addressing specific aspects of patient profiles and treatment responses. Our goal is to integrate these features and calculate clustering consistency throughout the process, improving patient adherence step by step. For instance, their research, for example, showed how well a Hidden Markov Model (HMM) could identify patient clusters based on residual apnea-hypopnea index (rAHI) trajectories. This allowed the HMM to predict treatment outcomes and provide personalized actions. [10]. We also searched how surveys of feedback are integrated with data management for CPAP treatment and their consideration for designing special surveys to cover each type of intervention. Collecting their qualitative insights, we assess the perceived challenges and benefits, aiming to align these with patient feedback for a comprehensive care perspective [11],[12].Using quantitative data and qualitative patient feedback, we examine patterns of CPAP adherence. This twofold feature aids in recognizing patterns and advances our comprehension of the reasons behind the variations in patient adherence over time. [13],[14] ,[5].

In [5] shows how personalized homecare interventions demonstrated that patient profiling and adherence predictions could significantly enhance CPAP therapy outcomes by tailoring interventions based on patient needs. By combining these various methods, our study aims to establish a dynamic and adaptive framework that continuously refines patient profiles and interventions. Ensuring effective attention to the unique needs of each patient is crucial in improving treatment outcomes and adherence for OSA patients receiving CPAP therapy.

### B. Research Literature Methodology on Patient Profile

Patient identification was a focus of the first phase of our review. After carefully reviewing 66 articles on patient profiling, we came up with 20 articles. We examined their profiles to understand the perspective and experience of patients being treated for sleep disorders based on CPAP data and other OSA-related considerations. The proposed methodology includes personalized care and patient feedback, with an emphasis on both quantitative and qualitative data, to improve adherence and the patient therapy cycle. Diverse patient behaviors towards CPAP therapy, identified [12] necessitate tailored interventions. This research facilitates a deeper understanding of patient's needs by examining demographic, clinical, and psychological predictors of adherence. Crucially, the [15]emphasizes the importance of understanding and predicting patient adherence through qualitative insights, enabling healthcare providers to preemptively tailor interventions. In their work [16] They introduced the CTAP-CPAP framework, which utilizes machine learning models to predict the risk of therapy abandonment. Their study used classifiers such as Random Forest and Support Vector Machines to identify patients at risk. This approach offers a data-driven method for real-time patient monitoring and intervention.

Our study integrates qualitative patient feedback (monitoring survey, medical survey) with quantitative CPAP data to enhance personalization and improve the clustering method. By clustering patients based on CPAP usage patterns and feedback from surveys, we tailor interventions more effectively to improve adherence rates. Recent research has underscored the value of using cluster analysis to identify distinct patient groups based on therapy adherence patterns, which can inform more personalized and effective intervention strategies[17]. Furthermore, the effectiveness of tailored interventions based on sociological determinants has been emphasized, suggesting that a comprehensive approach that includes these factors can significantly improve adherence[18]. By leveraging these insights, our method incorporates a continuous evaluation cycle, ensuring treatment strategies evolve based on patient input, continuously refining patient profiles and intervention strategies, and ensuring that treatment plans are dynamically adjusted to meet the evolving needs of patients which leads to a more patient-centric care model.

### C. Research Literature Methodology on Sleep Therapy Survey

We went through the distinct stages of considering sleep therapy articles, after carefully screening 430 articles on sleep therapy surveys, we selected 30 that best aligned by focusing on those, that provided the most effective insights into the intricate relationship between sleep surveys and therapy. We refined our research based on information gathered from five main categories of medical sleep surveys, namely sleep assessment, fatigue, anxiety and depression, daytime sleepiness, and general health status. We selected six surveys that were most relevant and informative in these categories. The first sleep assessment category survey is the Pittsburgh Sleep Quality Index (PSQI)[19], which contains 24 items on a 4-point scale. We have evaluated various aspects of sleep, including sleep quality, duration, efficiency, disturbances, medication use, and daytime dysfunction. The Insomnia Severity Index[20] is a second survey in the same category. Sleep dissatisfaction, daytime disturbances, the ability of sleep problems to be noticed by others, and the degree of anxiety caused by insomnia are measured in this survey. It has seven items on an Epworth Sleepiness Scale[21], which measures a person's level of sleepiness in daily life, his or her risk of

falling asleep in eight different situations, and its duration. There are eight items on a four-point scale in the ESS survey. The next category is Fatigue, and for this category, the chosen survey is the Fatigue Severity Scale (FSS)[22]. The link between the presence of anxiety and depressive symptoms in newly diagnosed OSA patients and the link between psychological symptoms and acceptance or adherence to CPAP after one year. One of the categories for using medical sleep surveys is assessing general health status in the medical Outcomes. The selected survey for this purpose is Short Form-36 (SF-36)[23], which assesses eight health concepts including limitations in physical and social activities due to health problems, bodily pain, mental health, vitality, and general health perceptions. The final type of medical sleep survey related to anxiety and depression is the Beck Depression Inventory (BDI)[24]. The focus has mainly been on detecting depression or depressive thoughts. The Pittsburgh Sleep Quality Index has proven to be a valuable tool in the field of sleep treatment. It has provided valuable insights into the practical application of these surveys in patient treatment, including the duration for which they were used.

## III. PATIENT ADHERENCE REINFORCEMENT

### A. Methodological Approaches

Our study builds upon the findings of their achievement [3], emphasizing the critical importance of addressing pre-treatment adherence risks through personalized interventions. This approach involves meticulous patient profiling and thorough analysis of CPAP monitoring and medical sleep survey data. Our methodology, as illustrated in Fig.1, initiates with tailored interventions categorized by CPAP data. We promptly seek patient feedback on the effectiveness of secondary interventions, facilitating a systematic and adaptive approach that integrates quantitative data with qualitative input to refine adherence strategies effectively. To improve patient adherence to CPAP therapy, we implement a multi-faceted approach that integrates CPAP monitoring, intervention monitoring surveys, and medical surveys.

This methodology aims to create a comprehensive and personalized treatment plan by considering specific aspects of patient profiles and treatment responses. Our goal is to integrate these features and calculate cluster consistency throughout the process, improving patient adherence step by step. Our patient empowerment methodology is a dynamic, iterative process catered to each patient's changing needs. To enable targeted interventions, patients are first grouped according to their characteristics and adherence to treatment. Additionally, drawing from their finding[25], our study advocates for qualitative interventions like motivational interviewing to enhance CPAP adherence. Empowerment is achieved through continuous feedback and adaptation, as we work collaboratively with patients to refine interventions and support their journey. If a patient demonstrates consistent adherence and positive outcomes across iterations, they are considered empowered.

This ensures that patient support remains ongoing and adaptable, with necessary adjustments to enhance therapy engagement and success. When CPAP data indicate improved compliance, we proceed with a medical sleep survey to further refine our approach based on detailed observations. Conversely, if insufficient improvement is evident, we pivot to reconfigure strategies. This iterative approach optimizes

CPAP therapy management by integrating ongoing patient feedback into tailored treatment plans and leveraging data-driven insights to enhance outcomes and adherence. Before initiating the four steps of our methodology, we acknowledge the continuous flow of new data from previous intervention assignments and evolving patient information. Our objective is to cluster patients for targeted interventions while preserving the interventions defined in prior research[3]. By assigning patients to specific profiles, we can refine our approach more effectively. Ultimately, we aim to gather comprehensive patient data to define new levels of intervention.

This process involves navigating through all data streams, as it is shown in Fig.2, we accumulate sufficient information to propose new interventions. Data fusion in our approach focuses on dynamically integrating all available data, emphasizing the most impactful features. We employ model merging as it shown in Fig.2, testing various models to achieve optimal performance. Initially, all data are included, allowing the model to select relevant features, with continuous updates to refine these choices as patient profiles evolve. Once a cluster is defined, we associate it with the appropriate action, whether it be a monitoring survey or a medical survey. By continuously adapting interventions to evolving patient profiles, iterative refinement ensures accurate prediction of patient adherence and outcomes, ensuring interventions remain relevant and tailored dynamic integration improves patient empowerment and treatment effectiveness.

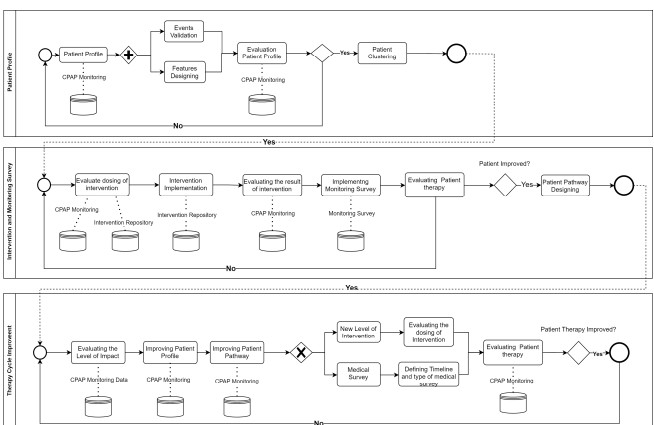

Fig. 1. Therapy Cycle

### a. Initial Clustering with CPAP Monitoring Data

Our initial clustering efforts revealed that the current data quality and quantity were insufficient for comprehensive analysis. Recognizing this, we focused on refining our clustering approach with new data. Patients undergo different types of surveys based on their needs and treatment phase. We prioritize clustering efforts to classify patients and define specific interventions, ensuring a dynamic process that allows continuous adjustment and personalization of care. Central to their strategy was the enhancement of patient-specific interventions and the exploration of how improved patient clustering could enhance CPAP adherence and compliance. They stratified patients based on CPAP monitoring data, identifying significant "events" in usage timelines such as variations in sleep metrics, adherence patterns, or usage habits. These events provided crucial insights into patient behaviors, guiding the adaptation of personalized care approaches for better treatment adherence and outcomes, and were essential for forming patient clusters. This process aligns

with the first stream in the Fig.1 and Fig.2, as shown in the models.

They started by analyzing continuous data from CPAP devices, including usage hours, leak levels, effective pressure, and residual Apnea-Hypopnea Index (AHI)[3]. These objective metrics are crucial for understanding patient adherence patterns, they segmented patients to identify stable clusters that accurately represented patient profiles. Initially, patients were categorized into three main adherence categories: Adherent, Non-Adherent, and Attempters, as they refined these clusters, they transitioned from three classifications of adherence to six detailed profiles: Adherent - Low Risk, Adherent - High Risk, Attempters - High Priority, Attempters - Low Priority, Non-Adherent, and Unknown, the first level of improvement involved refining these clusters until they stabilized, indicating that they had captured the most relevant patterns in the data[3].

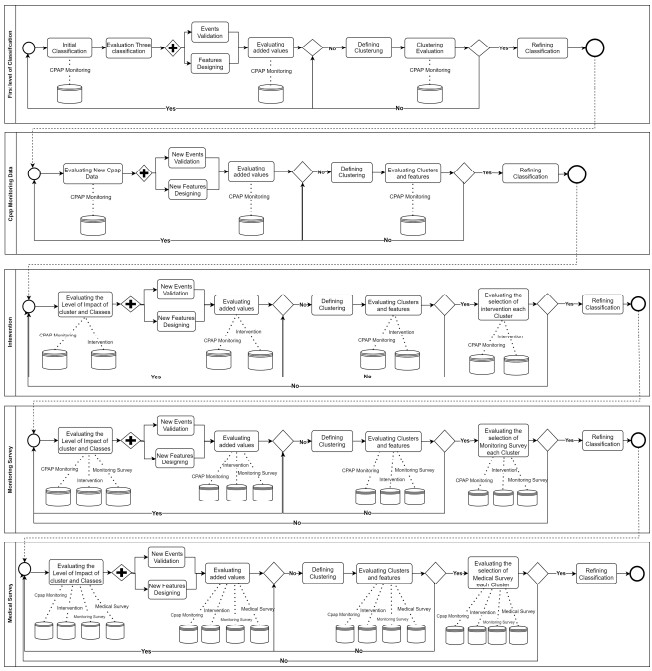

Fig. 2. Multi-Faceted Approach for CPAP Therapy Management

### b. New Clustering Loop in the first iteration with CPAP Monitoring Data

In the second iteration in Fig.2, we consider all features and select the best ones. We analyze data variability for each feature, ensuring the set of features used to constitute the model is clean and useful. Most of the data initially might not be well-formatted, impacting the model negatively. During model creation, some features may have less impact than others. These features are kept inside the model to observe their variability over time. As the data varies, these features might become significant, so they are not eliminated but monitored.

### c. Integration of Interventions' Data

Once the initial clusters from CPAP monitoring and survey data are stable, tailored interventions are implemented based on these clusters as shown in the third stream of Fig.2, Specific adjustments and support measures are designed to improve adherence. After each intervention, the data is revisited to add new features and observe their impact. If the number of clusters remains unchanged, it indicates optimal interventions have been found. The third phase focuses on

aligning interventions with specific patient profiles by associating clusters with corresponding profiles and analyzing the relationships between them. This allows for more accurately tailored interventions which are tested by altering data and observing their confirmation through monitoring surveys in the fourth stream, ensuring a stable and effective connection between interventions and clusters. The process is continuously improved by calculating and controlling the stability of information provided to patients, refining the approach step by step to enhance adherence and outcomes.

### d. Integration of Monitoring Survey Data

Following the intervention phase, we integrate data from regular monitoring surveys in the fourth stream of Fig.2. These surveys address patient experiences, device comfort, side effects, and adherence challenges, providing qualitative insights that complement the quantitative CPAP monitoring data. This step further enriches patient profiles and helps define a loop where we evaluate the impact of adding new features on clustering consistency. By connecting data from intervention monitoring surveys, we assume a better understanding of our patients. This integration proves that adding new features can improve patient profiles, clustering, and intervention selection. After integrating survey data and refining clusters, we perform a new classification to ensure that the patient profiles are up-to-date and accurately reflect their adherence patterns.

### e. Integration of Medical Survey Data

To refine care optimization, we leverage medical sleep surveys to adjust patient clustering and care strategies based on their impact on sleep cycles and treatment acceptance as defined in the last stream of Fig.2. Our approach is validated by comparing adherence patterns with and without survey feedback, which helps guide adjustments to ensure responsive and efficient CPAP therapy. Additionally, it helps us obtain sufficient results to compare each data stream, enabling us to assess the effectiveness of our approach and make informed adjustments for better patient outcomes. With stable clusters from CPAP monitoring, intervention data, and survey data, we expand our data sources to include comprehensive medical sleep surveys. These surveys provide detailed assessments of sleep quality and overall health. To make sure that patient profiles are current and appropriately reflect their adherence patterns, we perform a new classification after integrating medical survey data and fine-tuning clusters. By assessing the impact of these new data streams on cluster consistency, we improve our understanding of patient health and adherence behavior and enable us to dynamically adjust therapeutic strategies to improve adherence and ensure more targeted and effective interventions.

## IV. IMPLEMENTATION

To enhance the effectiveness of CPAP therapy for obstructive sleep apnea (OSA), we have implemented a series of detailed methodologies aimed at improving patient adherence and optimizing care. These steps are designed to integrate real-time data, patient feedback, advanced analytics, clustering, and intervention personalization. To simplify our approach for broader adoption, we defined it in three main steps: understanding patient needs, delivering targeted interventions, and collecting feedback. This framework is intuitive and focuses on key outcomes like increasing adherence and CPAP usage. First, we gather and analyze patient data to understand their specific needs. Next, tailored

interventions are delivered based on these insights. Finally, feedback is collected to evaluate the effectiveness of interventions, allowing for adjustments. This clear, modular design supports continuous learning and improvement, making the system accessible and effective across various settings.

Our methodology balances simplicity and necessary complexity through a modular design that supports continuous enhancement of patient care. While the approach is streamlined, some complexity is maintained to ensure robustness, adaptability, and responsiveness in real-world applications. For scaling in broader healthcare settings, including applications like oxygen therapy, we plan to validate the methodology's adaptability and effectiveness through clinical trials involving at least 70 patients. These trials will assess how personalized interventions improve CPAP adherence and empower patients, with adjustments made to align treatment with individual needs and real-life usage patterns.

To secure patient data, we adhere to ISO/IEC 27001 for information security management and ISO/IEC 27002 for security controls, aligned with GDPR. Data protection includes AES-256 encryption at rest, TLS/SSL encryption in transit, and anonymization techniques to safeguard personal identifiers while balancing privacy and utility. Access is restricted using role-based access controls (RBAC) and multi-factor authentication (MFA), ensuring data is handled only by authorized personnel. Regular data integrity checks in line with ISO/IEC 27001 maintain consistency and prevent tampering. For AI models, we follow ISO/IEC 23053 to secure the AI lifecycle, from development to deployment, preventing unauthorized changes and ensuring reliable clinical application. ISO/IEC 24029 ensures AI model trustworthiness by enhancing transparency, robustness, and accountability, making AI systems reliable and interpretable for healthcare providers. These standards guide security measures across the AI lifecycle, addressing vulnerabilities at every stage to uphold compliance and enhance patient care and data protection.

a. *Comprehensive Data Analysis and Clustering Methodology for CPAP Therapy Optimization in Linde Homecare*

Our study analyzed Linde's monitoring survey data and CPAP monitoring data, incorporating usage data from devices such as Philips, ResMed, Sefam, and Lowenstein. Over 10,000 raw data records were collected, including data from 570 patients, which contain 28% female and 71% male, providing extensive coverage of patient behaviors. Data collection was enhanced by technicians who completed questionnaires during personal visits, mobile apps, SMS, or calls. Tools such as NumPy, Pandas, Matplotlib, and Seaborn facilitated data extraction, formatting, and trend identification. Contingency tables were employed to examine relationships between monitoring survey responses and patients, identifying which questionnaires had the highest completion rates per intervention type.

To ensure the quality and consistency of the data, several preprocessing techniques were applied. SimpleImputer handled missing values by imputing with the mean or the most frequent value. LabelEncoder encoded categorical features as numerical values, while StandardScaler standardized features by removing the mean and scaling to unit variance. Additionally, ColumnTransformer was used to apply different preprocessing steps to various columns, ensuring that the data was well-prepared for analysis. By integrating these robust data engineering methodologies with statistical methods, we derived significant insights from the refined data. This comprehensive preprocessing ensured that our analysis was based on high-quality, well-structured data, enabling us to identify key trends and relationships within the dataset. Data selection criteria were designed to integrate various patient-related data types for creating AI models aimed at personalizing OSA treatment. This included CPAP monitoring data (e.g., UsageDuration, MaskPressure), patient demographics (e.g., Age, Gender, BMI), and patient-specific intervention records (e.g., InterventionType, OutcomeMetric). Additionally, risk factor data and responses from structured questionnaires were included, combining categorical and numerical identifiers to provide a comprehensive dataset. Quality assurance was maintained by using CPAP machines certified for accuracy, with strict calibration and verification standards. Data, such as UsageDuration and AHI, was anonymized and securely transmitted to servers for analysis. Patient feedback gathered via structured monitoring and medical surveys provided subjective insights, complementing the machine data. Key steps included secure transmission, data cleaning, standardization, and cross-referencing between machine logs and patient reports to ensure consistency and detect non-adherence early.

I defined several scenarios for our clustering analysis in Fig.3: CM for CPAP monitoring, CMPF for CPAP monitoring with patient features, CMIPF for CPAP monitoring with improved patient features, ICMPF for intervention and CPAP monitoring with patient features, ICMIPF for intervention with CPAP monitoring and improved patient features, MSICMPF for monitoring survey and intervention with CPAP monitoring and patient features and MSICMIPF for monitoring survey and intervention with CPAP monitoring and improved patient features. We began by clustering the CPAP monitoring data using KMeans. The optimal number of clusters was determined using KneeLocator with the Elbow method, resulting in five clusters and a silhouette score of 0.4026. Key metrics such as usage hours, leak levels, effective pressure, and residual Apnea-Hypopnea Index (AHI) were analyzed to identify stable clusters representing patient profiles.

Initial classifications included Adherent, Non-Adherent, and Attempters, which were refined into six detailed profiles for better accuracy. To enhance the clustering and classification process, patient features were incorporated. This step resulted in an optimal clustering solution with three clusters and a silhouette score of 0.9850. Significant features like Risk Factor Value and Description played a crucial role in this enhanced clustering. Once the clusters from CPAP monitoring and patient features were stable, tailored interventions were implemented based on these clusters. After each intervention, data was revisited to add new features and observe their impact. The optimal number of clusters remained at three, indicating the interventions were well-organized. This phase ensured interventions were aligned accurately with patient profiles, leading to more effective and personalized care. Finally, monitoring survey data was integrated to further refine clustering and intervention strategies. This step involved analyzing qualitative data from patient experiences, device comfort, side effects, and adherence challenges. The optimal clustering solution

involved six clusters with a silhouette score of 0.4214. Significant features impacting clustering included Postal Code, Service ID, Reference Date, Status, and Question ID. This enhanced result allowed us to identify key trends and relationships within the dataset, facilitating more effective and personalized care strategies. By continuously integrating various data sources and refining clustering methodologies, we were able to achieve a more accurate representation of patient profiles, ultimately improving patient adherence and optimizing therapy outcomes for obstructive sleep apnea.

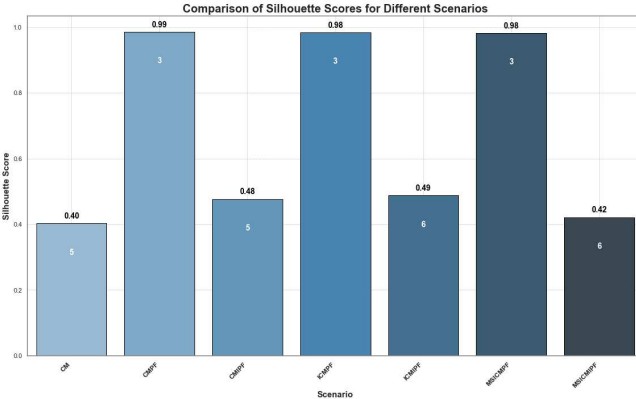

Fig. 3. clustering approaches

### b. Medical Sleep Survey

Our study embraced the data of real patient response patterns to fully comprehend patient behavior within the framework of CPAP therapy for obstructive sleep apnea (OSA). This novel method ensured the highest degree of dataset integrity through stringent cleaning and validation procedures. Sophisticated data analytics techniques, including correlation analysis, were employed to find relationships between different survey responses. We analyzed the findings from the medical sleep survey to assess the relationship between patient adherence to CPAP therapy for OSA and feedback from sleep therapy surveys. Drawing on important studies like [25] and [4], our research underscores how patient feedback significantly enhances treatment compliance. Our goal is to demonstrate how the success of CPAP therapy is directly impacted by integrating insights from sleep surveys, highlighting the vital role that patient-centered feedback plays in improving adherence outcomes. Through this structured approach, we extracted meaningful insights from the sleep survey data, aiding our understanding of the factors influencing patient responses. By utilizing correlation heatmaps, we examined relationships within survey sub-categories.

The SF-36 survey encompasses various dimensions of health and well-being, including 'Physical Functioning', 'Role Limitation due to Physical Health', 'Role Limitation due to Emotional Problems', 'Energy/Fatigue', 'Emotional Well-Being', 'Social Functioning', 'Pain', and 'General Health'. These facets provide a comprehensive view of patient health, which we analyze using correlation heatmaps to visualize associations between each pair of health dimensions within the survey. As shown in Fig.4. High correlations were observed between Emotional Well-being and Social Functioning (r = 1), indicating strong associations between these aspects of health. Conversely, low correlations were found between Pain and Social Functioning (r = 0.0027).

Significant correlations were identified in PSQI sub-categories between Sleep Duration and Habitual Sleep Efficiency (r=0.47), indicating these variables are closely related. In contrast, correlations between the Use of Sleeping Medications and Sleep Duration (r=0.15) or Daytime Dysfunction and Sleep Latency (r=0.25) were lower, with weaker associations between these variables. This comprehensive analysis helps us tailor treatment plans according to individual patient needs, preferences, and behavioral patterns. By understanding these connections, we can pinpoint factors influencing sleep health and guide interventions effectively. Our approach examines the interplay between patient-reported data and adherence outcomes, underscoring the utility of medical sleep surveys in developing nuanced patient profiles.

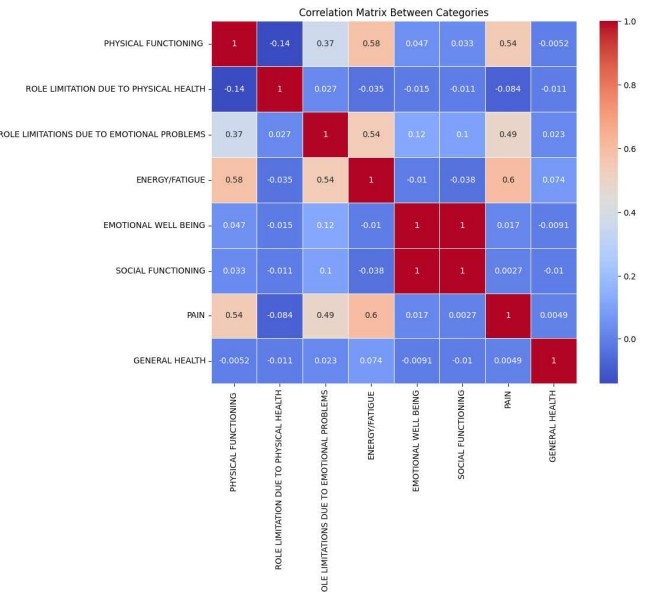

Fig. 4. Heatmap for all SF-36 sub-categories

### c. Case Study: Adherence Improvement in a well-adherent patient

To illustrate the adherence improvement capabilities in a well-adherent cluster (more than 6 hours), a detailed case study (patient 7583) While the single-patient case study served as an illustrative example demonstrating the methodology's application and effectiveness in a controlled setting, the methodology is not limited to this single instance. It was used to test and refine the approach before broader implementation. The methodology is applied to all patients, not just one, and the next phase will involve multi-center trials with a more diverse patient population to enhance the generalizability and robustness of insights. This study highlighted the variations in adherence and behavior through different phases, leveraging monitoring survey responses and monitoring data to provide comprehensive insights into patient interactions with their device January 18, 2024, a comprehensive questionnaire covering a range of topics, including device usage hours, leak levels, effective pressure measurements, residual AHI, mask comfort, and patient motivations, was administered as part of the intervention, which was specifically created for in-person patient visits. Before the intervention, the median CPAP usage was 6.37 hours, which increased to 7.86 hours post-intervention, marking an improvement of 1.49 hours. The

Apnea-Hypopnea Index (AHI) also showed positive changes, with a median AHI decreasing from 2 to 1.8 post-intervention, indicating a reduction in sleep apnea events. This intervention led to quantifiable improvements in CPAP usage and patient engagement, confirming the efficacy of our tailored approach. By integrating real-time monitoring data with in-depth patient feedback, our approach offers a novel methodology for enhancing treatment outcomes in obstructive sleep apnea (OSA) management, setting a new standard for patient-centered care, and emphasizing the effectiveness of tailored interventions in improving CPAP therapy adherence and compliance. In our study, we analyzed the impact of interventions on the monitoring outcomes of Continuous Positive Airway Pressure (CPAP) therapy, considering the interplay between CPAP data collection and intervention timings. This analysis acknowledged the diversity in patient experiences, highlighted by varied responses to monitoring surveys and interventions. Patient 7583's experience supports our expectation that customized interventions lead to quantifiable improvements in CPAP usage and patient engagement.

TABLE I. ANALYSIS

| Patient 7583 | | | |
|---|---|---|---|
| Usage Analysis | | AHI Analysis | |
| Usage | 6.75 | AHI | 2.2 |
| MeanUsage | 7.32 | Mean AHI | 1.1 |
| Min Usage | 0 | Min AHI | 0 |
| Max Use | 10.3 | Max AHI | 4.7 |
| STD | 2.27 | STD | 0.98 |
| Median Before Intervention | 6.37 | Median Before Intervention | 2 |
| Median After Intervention | 7.86 | Median After Intervention | 1.8 |
| Change Usage | 1.49 | Change AHI | -0.2 |
| RangeScore-Usage | 10.03 | RangeScore-AHI | 4.1 |

The case study of patient 7583 demonstrates how tailored interventions can lead to significant improvements in patient experience and adherence and the importance of understanding how patients interact with their CPAP device during stages of their procedure. Our methodology validates the effectiveness of the interventions offered, leading to a more successful patient-centered care model, and ensuring that interventions are effective and tailored to the specific needs of each patient, thereby improving patient adherence and compliance, and also strengthening the overall care model.

*d. Validation Results*

Analysis of 7,618 entries showed a 2.4-fold increase in device usage per intervention and questionnaire completion, with consistent improvements across a diverse patient population. This affirmed the scalability and reliability of Linde's monitoring and survey systems in optimizing CPAP therapy for obstructive sleep apnea. By continuously refining clustering and intervention strategies, we set a new standard for patient-centered care, emphasizing tailored interventions to improve treatment outcomes. Our study also focused on the impact of interventions on CPAP therapy outcomes Patient 7583 showed increased engagement and CPAP usage, alongside a reduction in the apnea-hypopnea index (AHI), demonstrating that tailored interventions can significantly improve adherence in obstructive sleep apnea management. This approach integrates real-time monitoring with patient feedback to create a dynamic and adaptive framework for optimizing the treatment of obstructive sleep apnea. By continuously evaluating and adjusting interventions based on patient data, we personalize care to ensure relevance and effectiveness. The adaptive cycle involves delivering interventions, assessing their impact, and incorporating feedback to refine the approach continuously. Key metrics, including intervention consumption, cost-effectiveness, treatment quality, and assessments of patient well-being (such as SF-36 for quality of life, PSQI for sleep quality, and ESS for daytime sleepiness), validate the effectiveness of this approach. These metrics allow for ongoing adjustments to interventions, enhancing adherence and overall therapy outcomes. By maintaining this adaptive cycle, we tailor interventions to each patient's specific needs and predict future treatment steps, thereby optimizing personalized care and improving treatment outcomes in obstructive sleep apnea.

## V. CONCLUSION

We have demonstrated that managing Obstructive Sleep Apnea (OSA) can be significantly enhanced through individualized treatment plans that integrate patient feedback with CPAP monitoring data. By refining patient clustering from three to six detailed profiles based on usage patterns, we developed targeted interventions that significantly improved adherence and treatment outcomes. This iterative and data-driven approach, supported by continuous feedback from interventions and surveys, allowed for dynamic adjustments to therapeutic strategies, ensuring personalized and effective care. The integration of diverse data sources enabled us to maintain up-to-date and accurate patient profiles that reflect adherence patterns. This comprehensive, patient-centered approach highlights the critical role of tailored care in OSA management and paves the way for more effective and responsive care models, ultimately enhancing patient outcomes and engagement.

## VI. RESULT

Our study demonstrated a 2.4-fold increase in CPAP device usage following tailored interventions, confirming the effectiveness of personalized strategies in enhancing patient engagement and adherence. Higher response rates to monitoring surveys indicated improved patient engagement, which is crucial for successful CPAP therapy. A case study of patient 7583 showed a median CPAP usage increase from 6.37 to 7.86 hours and a reduction in the Apnea-Hypopnea Index (AHI) from 2 to 1.8, further validating our approach. By refining the clustering approach from three main adherence categories to six detailed profiles, we ensured more accurate patient classification and targeted interventions. The integration of CPAP monitoring data, intervention outcomes, regular surveys, and medical sleep surveys allowed for consistency and precision in patient clustering. Analysis of 7,618 entries revealed consistent improvements across a diverse patient population, demonstrating the scalability and reliability of our system. These findings highlight the potential of personalized, feedback-driven care models to improve OSA management, adherence, and overall patient health, advancing toward a more effective, patient-centered care approach.

### ACKNOWLEDGMENT

This paper presents results that are developed in collaboration between the Linde Homecare France company and the University Lumiere Lyon 2, DISP Lab. This research is established under a CIFRE contract (2023/1249). The content of this paper reflects an R&D initiative promoted by

Linde Homecare France. Responsibility for the information and views expressed in this paper lies entirely with the authors.

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
