# OpenReview forum: "CPAP Adherence Improvement for OSA Patients through Integrated Feedback Systems"
_IEEE.org/EMBS/BHI/2024/Conference — IEEE BHI'24_

### Official Review · Reviewer_4ZM5 · 2024-07-28
**Review for the Article: "CPAP Adherence Improvement for OSA Patients through Integrated Feedback Systems"**

**Overall Rating:** 7
**Confidence:** 4

**Other Quality Metrics:**

Clarity of Writing: Good
Clinical Significance: Great
Methodological Novelty: Excellent
Experiments and Results: Good

**Questions For The Authors:**

1.- How do you validate the effectiveness of the patient clustering and tailored interventions over time? Are there specific metrics or benchmarks used to measure success?
2.- What specific security measures have been implemented to protect patient data during the integration of CPAP machine monitoring and patient feedback?
3.- How do you plan to scale this methodology for broader use in different healthcare settings? Are there plans for pilot studies or larger clinical trials?

**Strengths:**

1.- The paper combines qualitative feedback with quantitative data to tailor interventions for CPAP therapy, addressing a significant challenge in OSA treatment.
2.- The approach involves advanced patient clustering, continuous data integration, and iterative refinement of treatment plans, making the methodology robust and adaptive.
3.- By focusing on personalized interventions and patient-centered care, the study aims to enhance treatment efficacy and patient adherence, which is crucial for the success of CPAP therapy.

**Summary Of The Paper:**

The paper presents an innovative strategy to enhance adherence to Continuous Positive Airway Pressure (CPAP) therapy for patients with Obstructive Sleep Apnea (OSA). It emphasizes the integration of qualitative patient feedback with quantitative CPAP machine data to create personalized treatment plans. The study aims to improve CPAP adherence through advanced patient clustering, dynamic data updates, and tailored interventions.

**Weaknesses:**

1.- The detailed case study is based on a single patient, which may limit the generalizability of the findings. Expanding the sample size would provide more robust insights.
2.- The multi-layered approach, while comprehensive, is complex and may require significant resources for implementation in real-world settings. Simplifying the methodology could enhance its practical applicability.
3.- The paper does not provide a thorough analysis of data security measures necessary for protecting patient information, which is critical when dealing with sensitive health data.

---

### Official Review · Reviewer_fijn · 2024-07-30
**Overall this study is important, but the approach is complicated and not easy to implement. The results are appropriate but promising.**

**Overall Rating:** 6
**Confidence:** 5

**Other Quality Metrics:**

(a) Clarity of writing: great
(b) Clinical Significance:good (c) Methodological Novelty: good (d) Experiments and Results: great

**Questions For The Authors:**

Can the procedures be simplified?

**Strengths:**

A multi-layered approach has been developed to improve adherence and a patient- centered care model for individuals with OSA by continuously adapting and personalizing the care was developed.

**Summary Of The Paper:**

This study  aimed to Integrate feedback systems to improve adherence of  Continuous Positive Airway Pressure (CPAP) therapy for OSA patients.

**Weaknesses:**

The developed approaches or procedures  are completed and not easy to implement or adopted to other studies and the usage is limited.

---

### Official Review · Reviewer_yd8x · 2024-08-16
**CPAP Adherence Improvement for OSA Patients through Integrated Feedback Systems**

**Overall Rating:** 5
**Confidence:** 4

**Other Quality Metrics:**

Some spelling errors exists such as "Furthermore, This study[4] demonstrates that digital feedback, tailored to patient-specific..."

**Questions For The Authors:**

1) How the data are selected in terms of data quality assurance?
2) How the data are fused and merged?
3) How can you guarantee the effectiveness of your approach?

**Strengths:**

The proposed approach combines quantitative data from CPAP machine monitoring with qualitative data from several types of
surveys. Qualitative data for our study derived from two primary sources: first, interventions that are informed by monitoring survey that gathers patient feedback on their treatment experiences following each intervention; and second, Medical Sleep Surveys to evaluate broader issues related to sleep health and quality.

**Summary Of The Paper:**

This paper deals with Obstructive Sleep Apnea (OSA). In particular, it gives emphasis on patient empowerment through the integration of feedback management systems into sleep treatment.

**Weaknesses:**

The criterai for data selection is weak and not not well documented. The methodology used for patients; empowerment is rather weak.

---

### Decision · Program_Chairs · 2024-09-23

Accept